# Polyvinylpyrrolidone-Coated Catheters Decrease Astrocyte Adhesion and Improve Flow/Pressure Performance in an Invitro Model of Hydrocephalus

**DOI:** 10.3390/children10010018

**Published:** 2022-12-22

**Authors:** Leandro Castañeyra-Ruiz, Seunghyun Lee, Alvin Y. Chan, Vaibhavi Shah, Bianca Romero, Jenna Ledbetter, Michael Muhonen

**Affiliations:** 1CHOC Children’s Research Institute, and CHOC Neuroscience Institute, 1201 W. La Veta Avenue, Orange, CA 92868, USA; 2Neurosurgery Department, CHOC Children’s Hospital, 505 S Main St., Orange, CA 92868, USA

**Keywords:** polyvinylpyrrolidone, bioglide, catheter, flow/pressure performance, astrocytes, hydrocephalus

## Abstract

The leading cause of ventricular shunt failure in pediatric patients is proximal catheter occlusion. Here, we evaluate various types of shunt catheters to assess in vitro cellular adhesion and obstruction. The following four types of catheters were tested: (1) antibiotic- and barium-impregnated, (2) polyvinylpyrrolidone, (3) barium stripe, and (4) barium impregnated. Catheters were either seeded superficially with astrocyte cells to test cellular adhesion or inoculated with cultured astrocytes into the catheters to test catheter performance under obstruction conditions. Ventricular catheters were placed into a three-dimensional printed phantom ventricular replicating system through which artificial CSF was pumped. Differential pressure sensors were used to measure catheter performance. Polyvinylpyrrolidone catheters had the lowest median cell attachment compared to antibiotic-impregnated (18 cells), barium stripe (17 cells), and barium-impregnated (21.5 cells) catheters after culture (*p* < 0.01). In addition, polyvinylpyrrolidone catheters had significantly higher flow in the phantom ventricular system (0.12 mL/min) compared to the antibiotic coated (0.10 mL/min), barium stripe (0.02 mL/min) and barium-impregnated (0.08 mL/min; *p* < 0.01) catheters. Polyvinylpyrrolidone catheters showed less cellular adhesion and were least likely to be occluded by astrocyte cells. Our findings can help suggest patient-appropriate proximal ventricular catheters for clinical use.

## 1. Introduction

Hydrocephalus is commonly a dynamic process with enlargement of the brain ventricles due to cerebrospinal fluid (CSF) flow impairment. In the United States, it affects 80–125 children per 100,000 births [1,2]. Pediatric hydrocephalus is frequently accompanied by increased intracranial pressure, brain structure compression, and neurological impairments. The primary treatment is ventriculoperitoneal shunting [3,4,5]. However, CSF shunts fail at a high rate [6,7], with up to 90% failing within 15 years [4]. Over half of all pediatric shunt failures are due to ventricular catheter tissue blockage with astrocytes being one of the most prevalent cells adhering to ventricular catheters [8,9,10].

Efforts to prevent shunt blockage have focused on modifying cell adhesion properties and immuno-activating the ventricular catheter [11,12]. However, these methods may not successfully mitigate the immune reaction cascade associated with catheter implantation, which could lead to a cellular blockage [12]. Even with the understanding that astrocytes and macrophages are the main contributors [9], the precise molecular processes underlying catheter blockage are poorly understood. This study assessed the occlusion probability of different commercially available catheters in our validated catheter assay system [13] to help neurosurgeons make informed decisions when selecting proximal shunt catheters.

## 2. Materials and Methods

### 2.1. Cell Adhesion to Catheters

The distal portions of ventricular catheters (Table 1) were cut under sterile conditions into 0.5 cm pieces and placed in 2 × 24-well plates (12 pieces per catheter type). A human astrocyte culture kit (N7805-200, Gibco) was used to test cellular adhesion to the different catheter materials (Table 1). Astrocytes were thawed at 37.0 °C for 2 min, centrifuged at 290× *g* for 5 min, and resuspended at 105 cells per mL in astrocyte medium (N7805-200 kit, Gibco; DMEM, 1% N-2 Supplement, 10% FBS). Finally, 250 µL of the mixture was seeded onto 24-well plates holding the catheters. After 24 h, the cells were processed for immunocytochemistry to quantify cellular adhesion.

### 2.2. Catheter Occlusion

The proximal portions of ventricular catheters were cut under sterile conditions (approximately 4 cm). Astrocytes were cultured in a VitroGel 3D TDS (TWG001, The Well Bioscience) as a proxy for the extracellular matrix. VitroGel 3D TDS was mixed with dilution solution in a 1:5 proportion. Cells were thawed at 37 °C for 2 min and centrifuged at 290× *g* for 5 min. Cells were resuspended at 106 cells/mL in astrocyte medium. Astrocytes were mixed with VitroGel 3D TDS 1:5 to obtain a final concentration of 105 cells/mL. Finally, 30 µL of the mixture was inoculated into the ventricular catheters to fill the proximal end of the ventricular catheter internally. After 15 min of VitroGel 3D TDS polymerization, the catheters were placed into a 100 mm Petri dish with astrocyte medium containing epithelial growth factor (PHG0314, Thermo Fisher Scientific) at 20 ng/mL. Catheters were incubated under standard conditions (37 °C, 5% CO_2_) to support astrocytic growth. The media was renewed every 48 h. After 7 days, catheters were inserted into the 3D brain phantom to test catheter obstruction (Figure 1).

### 2.3. 3D Brain Phantom

The brain phantom was printed by a 3D printer (Form 3B, Formlabs) using flexible resin (Elastic 50A, Formlabs). The phantom had a volume of 230 mL to mimic the brain of a child afflicted with hydrocephalus, as previously described by our research group [13].

### 2.4. Assessment of Catheter Pressure and Flow

Benchtop testing was performed to measure the hydrodynamic characteristics of the catheters in terms of CSF flow rate and differential pressure. Artificial CSF flow rate was assessed as it exited the catheter. The differential pressure was determined as the difference between the recorded pressure within the phantom and the pressure exiting the phantom through the proximal catheter (Figure 2). The benchtop setup consisted of a 3D-printed brain phantom, syringe pump (Fusion 200-X, Chemyx Inc., Stafford, TX, USA), pressure/flow rate sensors (PX26-001DV, Omega, Norwalk, CT, USA), and customized electric circuits for data collection. The sample ventricular catheter was placed into the phantom to reproduce the conditions of shunt failure due to obstruction. A syringe pump filled with artificial CSF (LRE-S-LSG-1000-1, EcocyteShop) was set to a flow rate of 0.33 mL/min, as verified with a calibrated scale. The resistive pressure sensors (PX26-001DV, Omega), which have 1 mm H_2_O resolution, were calibrated to customized settings for measuring differential pressure and flow rate [13,14]. All data were recorded by a data acquisition board (NI USB 6216, National Instruments, Austin, TI, USA) and SignalExpress software. A customized electric circuit used for signal amplification from the sensors consisted of a resistor (470 W) and op-amps (INA121PA, Texas Instruments, Dallas, TI, USA) for high signal amplification gain (~100) (Figure 2).

### 2.5. Immunocytochemistry

For cellular quantification, a standard immunocytochemical protocol was used [15]. The catheters were rinsed twice with phosphate-buffered saline containing 1% Triton (PBSt) to remove the remaining artificial CSF or media. Cells were then fixed with paraformaldehyde in PBS for 7 min. Next, the catheter and cells were rewashed with PBSt and permeabilized with 5% bovine serum albumin and 1% Triton X-100 for 1 h. Cells were incubated for 2 h with anti-GFAP (Abcam, ab7260) at 1:400 in PBSt, to label astrocytes. Cells were washed twice with PBSt and stained with secondary antibodies (Alexa Fluor 488, Thermo Fisher Scientific, A11034). Cells were washed twice in PBSt and stained with DAPI at 1:5000 in PBS for 5 min to label nuclei.

### 2.6. Image Analysis

Images of immunostained catheters were obtained using the fluorescence ECHO-Revolve microscope (Echo). ImageJ (NIH image software) was used to quantify total cells as previously described [16], in areas of 0.45 mm^2^ for the cell adhesion assay and 0.071 mm^2^ for the catheter obstruction assay.

### 2.7. Statistical Analysis

For cellular quantification, the Mann–Whitney U-test was used to compare values. Data are presented as median with data range (minimum to maximum and interquartile range). Differences were considered statistically significant at *p* < 0.05. For flow and pressure quantification, 20 values were obtained per second, and the experiments were carried out for 120 s. Data sets are expressed as mean ± SEM. The Friedman test was used to determine whether differences between groups were significant, with differences being considered statistically significant at *p* < 0.05. All analyses were conducted using Prism 9 (GraphPad Software).

## 3. Results

### 3.1. Cellular Adhesion

Astrocytes were seeded onto different commercial catheters to test the affinity of these cells to the catheters. The polyvinylpyrrolidone catheters showed significantly less median cell attachment (1 cell per 0.45 mm^2^) compared to the other catheters (barium-stripe 17, barium-impregnated 21.5, antibiotic-impregnated 18; *p* < 0.01) (Figure 3).

### 3.2. Catheter Performance under Obstructive Conditions

Astrocytes were cultured within ventricular catheters for 7 days to induce cellular ventricular catheter obstruction. The catheters were placed in the catheter performance system to test flow and pressure. Finally, the catheters were processed for immunocytochemistry to quantify cellular adhesion in the areas juxtaposed to catheter holes.

In the cellular-obstructed catheters, the differential pressure between the phantom and the catheter increased significantly in barium stripe catheters (9.53 ± 2.51 cm H_2_O, *p* < 0.001), in barium-impregnated catheters (3.72 ± 1.87 cm H_2_O, *p* < 0.001), and in antibiotic-impregnated catheters (2.36 ± 0.30 cm H_2_O, *p* < 0.001), compared with polyvinylpyrrolidone catheters (0.24 ± 0.10 cm H_2_O). Barium stripe catheters showed significantly increased pressure compared to antibiotic- and barium-impregnated catheters (*p* < 0.001). Inversely, flow rate was significantly increased in polyvinylpyrrolidone (0.105 ± 0.002 mL/min, *p* < 0.001), antibiotic-impregnated (0.097 ± 0.022 mL/min, *p* < 0.001) and barium-impregnated catheters (0.080 ± 0.009 mL/min, *p* < 0.001), when compared with barium stripe catheters (0.020 ± 0.022 mL/min) (Figure 4A,B). After catheter performance was evaluated, the number of cells adhered to the catheters’ holes was quantified. Polyvinylpyrrolidone catheters demonstrated significantly less cellular adherence (median 0 *p* < 0.01) when compared with the other catheter types studied (barium stripe 3.5, barium impregnated 2, antibiotic-impregnated 4) (Figure 3C,D).

## 4. Discussion

This study focused on testing astrocytic affinity for various commonly implanted ventricular catheters. Our results indicate that there is significantly less cellular attachment in the polyvinylpyrrolidone catheters, other in vitro studies reported decreased bacterial attachment in silicone tubing treated with polyvinylpyrrolidone [17].

Few in vitro models have been developed to elucidate catheter obstruction mechanisms, with most focusing on glial reaction or blood coagulation [10,18,19,20,21,22,23]. This study uses our previously described 3D-printed phantom ventricular replicating system [13] to test cellular-dependent obstruction after exposure to ex vivo astrocytes. We found that the polyvinylpyrrolidone catheters were less likely to become occluded with ex vivo astrocytes. This finding may suggest that polyvinylpyrrolidone catheters are less likely to have shunt failure from primary astrocyte catheter occlusion.

Astrocytes and reactive astrogliosis are highly associated with hydrocephalus cytopathology [24,25,26,27,28], contributing to proximal shunt catheter occlusion [9,10,11,12,29]. Our experiments with astrocytes indicate that polyvinylpyrrolidone catheters were the least likely to occlude due to cellular adhesion. This could have many implications, including decreasing the likelihood of proximal shunt failure resulting from astrocyte adhesion to the catheters.

When tested in our phantom ventricular replicating system, polyvinylpyrrolidone catheters demonstrated the least resistance with the greatest flow because the astrocytes did not attach to it and did not occlude the inflow holes. However, these catheters may be prone to disconnection. A retrospective review reported that of 466 patients implanted with polyvinylpyrrolidone proximal shunt catheters, 23 (5.6%) experienced shunt failure due to disconnection from the valve and migration of the proximal catheter [30]. While our experiments demonstrated that polyvinylpyrrolidone proximal shunt catheters might be the least likely to occlude from astrocytic growth, shunt failure may still occur for various other reasons.

### Limitations

There are several limitations to consider. First, there are many ways a shunt can fail, including valve damage, fracture or dislocation, and distal shunt occlusion, which were not tested in our system. Second, although astrocytes have been implicated as primary cellular drivers of proximal shunt occlusion, there are additional cell types or tissue (choroid plexus, microglial cells, etc.) and physiological drivers (blood clots, etc.) contributing to shunt occlusion that were not investigated in our study. It has been reported that lymphocytic inflammation is one of the main reasons for catheter obstruction, accounting for 29% of obstructed catheters [9]. This is probably associated with inflammatory-dependent lymphocyte infiltration through the blood–brain barrier. In the same study, choroid plexus obstruction accounted for 24% of the cases, glial tissue occlusion (mainly astrocytes) represented 24% of the obstructed catheters and foreign body giant cell reactions 5% of the cases [9,31].

In addition to the above limitations, the lack of other factors that affect cellular attachment in our ex vivo conditions, such as the influence of immune cells or inflammatory reactions may cause this work not to be reproducible in in vivo conditions.

## 5. Conclusions

Polyvinylpyrrolidone catheters demonstrated the lowest propensity for astrocytic occlusion during testing in our phantom ventricular replicating system, as they exhibited significantly lower cellular adhesion. Although polyvinylpyrrolidone catheters were the least likely to be occluded in our system, their patency performance in actual patients with shunted hydrocephalus is still unclear, and their use may be associated with unique complications.

## Figures and Tables

**Table 1 children-10-00018-t001:** Ventricular Catheters used in the study.

Product Description	Medtronic Catalog Number	Quantity
Ares, antibiotic- and barium-impregnated	91,101	5
Bioglide (polyvinylpyrrolidone)	91,503	5
Barium-striped	24,154	5
Barium-impregnated	41,101	5

**Figure 1 children-10-00018-f001:**
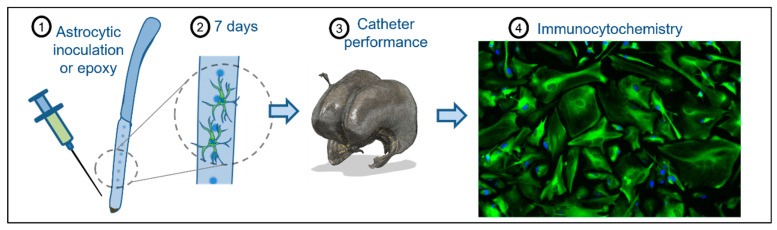
Schematic representation of the catheter occlusion procedure and testing. (1) Astrocytes were inoculated into the ventricular catheter using Vitrogel as an extracellular matrix for 7 days (2). Catheter performance was tested using the 3D brain phantom (3). Anti-GFAP (green) and DAPI (blue) immunohistochemistry staining was performed to quantify cellular attachment (4).

**Figure 2 children-10-00018-f002:**
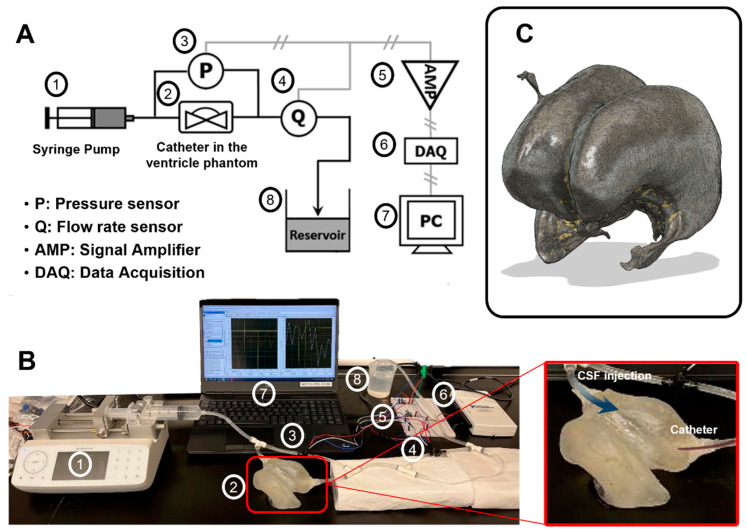
Catheter performance system. (**A**); Schematic of the experimental setup. A syringe pump (Fusion 200-X, Chemyx Inc.) injecting artificial CSF at 0.33 mL/min (1) into the 3D-printed ventricular phantom (2). Commercial resistance pressure sensors (PX26-001DV, Omega) were used to measure pressure and flow rate with customized calibrations (3 & 4). Electric voltage signals from the sensors were transmitted through the customized voltage amplifier (5) to the data acquisition board (NI USB 6216, National Instruments) (6) and recorded by SignalExpress software on a PC (7). (**B**); Photograph of the test setup indicating the different parts described in A (1-7). (**C**); 3D-printed ventricular phantom; 3D design layout. The phantom was designed by Fusion 360 software and fabricated by SLA (Stereolithography) 3D printer (Form 3B, Formlabs) using elastic resin (Elastic 50A, Formlabs).

**Figure 3 children-10-00018-f003:**
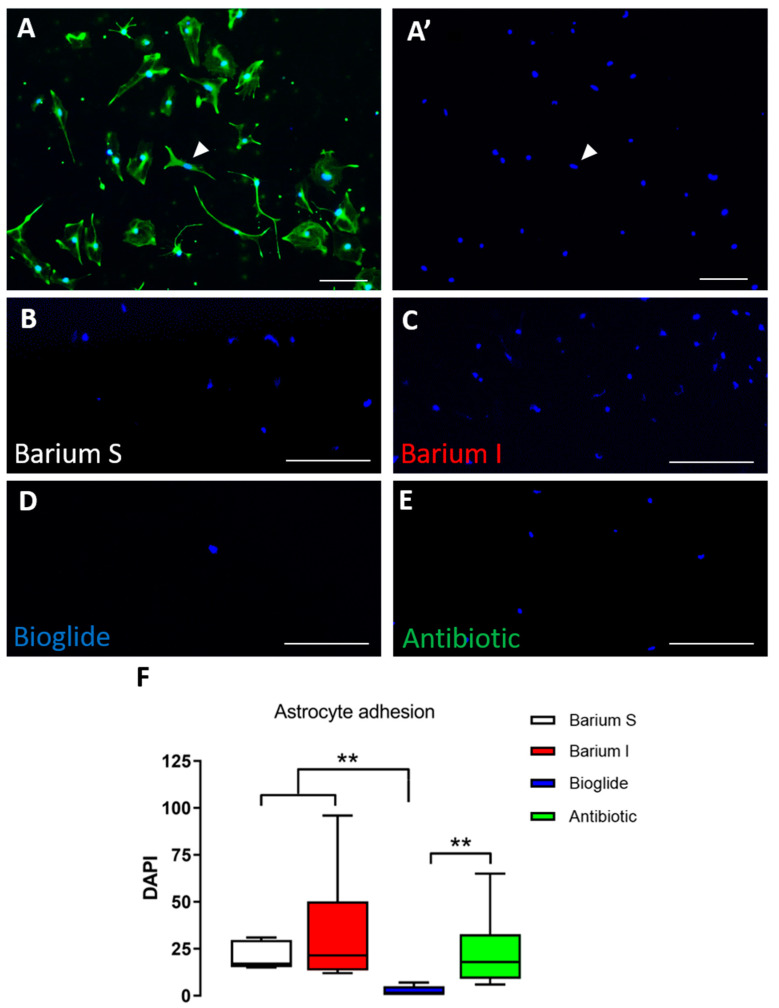
Astrocyte adhesion on the surfaces of commercial ventricular catheters. (**A**,**A’**); Immunocytochemistry against GFAP, a specific astrocyte marker, and DAPI (**A**) to label nuclei (arrowhead). Cellular adhesion was quantified using DAPI (**A’**). (**B**); Representative image of DAPI labeling after 24 h of cellular adhesion onto barium stripe catheters. (**C**); Representative image of DAPI after cellular adhesion onto barium-impregnated catheters. (**D**); Representative image of DAPI after cellular adhesion onto bioglide catheters. (**E**); Representative image of DAPI after cellular adhesion onto antibiotic-impregnated catheters. (**F**); Graphical representation of cellular adhesion in the different catheter types, indicating significantly less adhesion in Bioglide catheters (** *p* < 0.01). Data included *n* = 6 values per condition and were analyzed with the nonparametric Mann–Whitney U-test. Barium S = barium stripe; Barium I = barium impregnated; Bioglide = polyvinylpyrrolidone; Antibiotic = antibiotic-impregnated. Scale bars: 100 µm.

**Figure 4 children-10-00018-f004:**
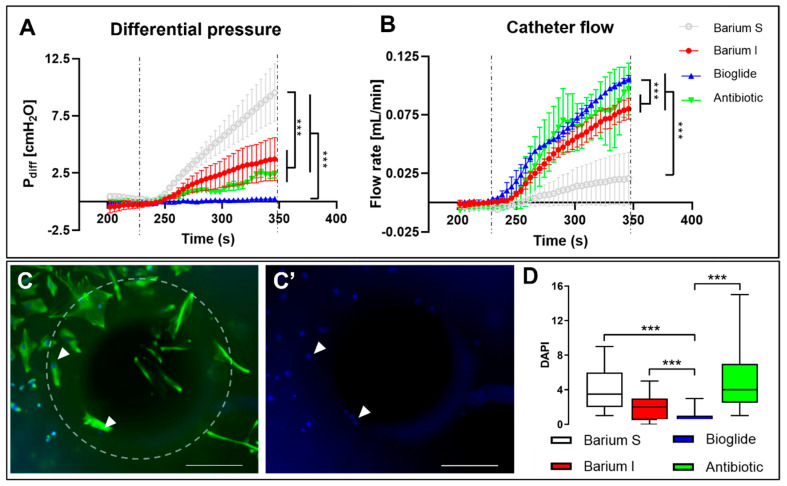
Catheter performance under obstructive conditions. (**A**); Graphic representation of intracranial-like pressure associated with astrocytic growth within commercial catheters. Significant pressure differences were found between all catheter types (*** *p* < 0.001), excluding the comparison between Barium I and antibiotic-impregnated catheters. Barium S demonstrated the highest pressure, and bioglide demonstrated the lowest pressure. (**B**); Graphic representation of flow associated with astrocytic growth into commercial catheters. Significant flow differences were found between all catheter types (*** *p* < 0.001) for the comparison between Barium I and antibiotic-impregnated catheters. Barium S exhibited the lowest flow, and Bioglide exhibited the highest flow (*** *p* < 0.001). Dashed lines indicate pump on and off time points. Data were obtained from 5 catheters per condition. Results were analyzed with the Friedman test. C & D; Immunocytochemistry against GFAP to label astrocytes and DAPI (**C**) to label nuclei (arrowhead). Cellular adhesion was quantified using DAPI (**C’**). (**D**); Graphical representation of cellular adhesion in all catheter types, indicating significantly less adhesion in Bioglide catheters (*** *p* < 0.001). Data included *n* = 20 values per condition and were analyzed with the nonparametric Mann–Whitney U-test. Barium S, barium stripe; Barium I, barium impregnated; Bioglide, Bioglide impregnated; Antibiotic, antibiotic-impregnated. Scale bars: 100 µm.

## Data Availability

All the data are available within the manuscript.

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
