# Peer review of "Polyvinylpyrrolidone-Coated Catheters Decrease Astrocyte Adhesion and Improve Flow/Pressure Performance in an Invitro Model of Hydrocephalus"

_children, 2022, doi:10.3390/children10010018_

Round 1
Reviewer 1 Report
The authors present a study of an ex vivo model of astrocyte adherence to 4 different ventricular shunt catheters. They show lower astrocyte adherence to the Bioglide catheters. They then go on to test flow through the astrocyte coated catheters in a synthetic brain model. Overall, there is better flow through the Bioglide catheters in their bench top system.
The science seems rigorous and the results are convincing. The largest limitation is the lack of recognition that the results from this ex vivo study may not be also true in an in vivo system. It is possible that the chemical coating on the catheter is inhibitory for the particular astrocyte from this culture experiment, but not for other astrocytes or for an in vivo model.
Reviewer 2 Report
This is an interesting study showing which surface with different treatment in clinical available shunts will give rise to a lesser degree of cell attachment related with the shunt failure due to shunt obstruction. The study uses astrocyte as a primary cause contributing to the obstruction of the shunt and demonstrated that among four models there is one that shows the minimal blockade due to astrocyte presence on the surface of the shunt material. This is a study that neurosurgeons and neuroscientist may want to take a look for a better shunt design and their utility in the clinic. While I enjoy the content of the study, as authors addressed in the limitation part of the discussion, there should be the cellular and non-cellular components leading to the blockade of the shunt other than astrocytes. The present reviewer hopes that the authors develop a more scientific hypothesis related with shunt failure in human clinical studies. For example, one can hypothesize that "shunt fails as the brain barrier dysfunctions with increasing age and that more numbers of T lymphocytes may infiltrate into the brain causing inflammatory state in the intraventricular and extraventricular space". Therefore, authors can test the hypothesis that it is not astrocytes, but cytotoxic (CD8+) T cells delivered through CSF and that attached to the surface of the shunt, which indicates the compromised glymphatic clearance. In that respect, why don't you add such a story or literature in your discussion and introduction (T-cell contribution to shunt failure)?
When you address that, the manuscript will be much more improved for many readers who might follow your paper.
